# Solving an Old Puzzle: Elucidation and Evaluation of the Binding Mode of Salvinorin A at the Kappa Opioid Receptor

**DOI:** 10.3390/molecules28020718

**Published:** 2023-01-11

**Authors:** Kristina Puls, Gerhard Wolber

**Affiliations:** Department of Biology, Chemistry and Pharmacy, Institute of Pharmacy, Freie Universität Berlin, Königin-Luise-Str. 2+4, 14195 Berlin, Germany

**Keywords:** GPCRs, kappa opioid receptor, Salvinorin A, natural products, docking, molecular dynamics simulations, dynophores

## Abstract

The natural product Salvinorin A (SalA) was the first nitrogen-lacking agonist discovered for the opioid receptors and exhibits high selectivity for the kappa opioid receptor (KOR) turning SalA into a promising analgesic to overcome the current opioid crisis. Since SalA’s suffers from poor pharmacokinetic properties, particularly the absence of gastrointestinal bioavailability, fast metabolic inactivation, and subsequent short duration of action, the rational design of new tailored analogs with improved clinical usability is highly desired. Despite being known for decades, the binding mode of SalA within the KOR remains elusive as several conflicting binding modes of SalA were proposed hindering the rational design of new analgesics. In this study, we rationally determined the binding mode of SalA to the active state KOR by in silico experiments (docking, molecular dynamics simulations, dynophores) in the context of all available mutagenesis studies and structure-activity relationship (SAR) data. To the best of our knowledge, this is the first comprehensive evaluation of SalA’s binding mode since the determination of the active state KOR crystal structure. SalA binds above the morphinan binding site with its furan pointing toward the intracellular core while the C2-acetoxy group is oriented toward the extracellular loop 2 (ECL2). SalA is solely stabilized within the binding pocket by hydrogen bonds (C210^ECL2^, Y312^7.35^, Y313^7.36^) and hydrophobic contacts (V118^2.63^, I139^3.33^, I294^6.55^, I316^7.39^). With the disruption of this interaction pattern or the establishment of additional interactions within the binding site, we were able to rationalize the experimental data for selected analogs. We surmise the C2-substituent interactions as important for SalA and its analogs to be experimentally active, albeit with moderate frequency within MD simulations of SalA. We further identified the non-conserved residues 2.63, 7.35, and 7.36 responsible for the KOR subtype selectivity of SalA. We are confident that the elucidation of the SalA binding mode will promote the understanding of KOR activation and facilitate the development of novel analgesics that are urgently needed.

## 1. Introduction

Proper pain management is an ongoing issue in medicine [1,2,3,4]. Efficient, strong, and safe analgesics for severe pain treatment are highly needed as currently available pain medications suffer from clinical drawbacks, like side effects, addiction, and overdose-related deaths [1,3,5]. Today, the most important drugs for severe pain treatment are opioids that mainly target the µ-opioid receptor (MOR) [5,6]. These drugs exhibit a high liability for drug abuse and elicit a myriad of side effects like respiratory depression, addiction, and constipation among others [5,7,8,9]. To overcome the current issue of insufficient severe pain medication the research focus shifts toward alternative targets [1]. The κ-opioid receptor (KOR) is a subtype of the opioid receptor (OR) family, belonging to the large class of membrane-embedded G protein-coupled receptors (GPCRs) [5]. Activation of the KOR provides strong analgesia without addiction and respiratory depression turning the KOR into a promising target for the development of new analgesics with an improved safety profile [6,10].

Nature hosts a plethora of diverse chemical scaffolds, many of which exhibit biological effects rendering natural products a promising source for the search for new drug candidates [11,12]. One such natural product, Salvinorin A (SalA, Figure 1), gained attention as an untypical, novel-scaffold KOR ligand [13,14,15]. SalA is a diterpene from the medicinal plant *Salvia divinorum* (Lamiaceae) endemic to Mexico [13,14,16]. *Salvia divinorum* was traditionally used by the Mazatecans for religious but also medicinal purposes like pain treatment, rheumatism, and inflammatory diseases [13,16]. In higher concentrations, SalA and *Salvia divinorum* products elicit strong hallucinogenic effects which led to the recreational use of SalA-containing products recently [13,14,16,17].

Due to its diterpene structure, SalA lacks the nitrogen atom typical for small molecules OR ligands, like morphine, fentanyl, and buprenorphine, which are mainly alkaloids [16,18]. An ionic interaction between the positively charged nitrogen of the ligand and the carboxylate of D^3.32^ (Superscripts denote Ballesteros-Weinstein numbering [19]) was believed to be crucial for OR affinity and activity of small molecule OR ligands thereby serving as an anchor point for the ligand [20,21,22,23]. SalA was the first nitrogen-lacking OR ligand discovered [14] underlying its unique character. Furthermore, SalA shows high selectivity for the KOR acting as a potent agonist without binding towards the MOR and the DOR (δ-opioid receptor) [24,25,26,27]. A broad screen of SalA against 50 transporters and receptors revealed no substantial activation besides its KOR activity [13,28]. Of note, SalA does not modulate the 5-HT_2A_-receptor which is commonly targeted by hallucinogens [13,27]. Nonetheless, Dopamin2 receptor modulation, allosteric MOR modulation, and allosteric modulation of cannabinoid type 1 receptors are discussed [13,14].

The potential of SalA as a new analgesic is not only supported by its high KOR selectivity, but also by its overall low toxicity with no severe adverse outcomes reported [13,16]. Despite its hallucinogenic character and its recreational usage SalA does not cause addiction [13,14] in contrast to typical opioids [5]. Nonetheless, reports about typical KOR-mediated side effects like anhedonia, locomotor impairment, and aversion caused by SalA but also its hallucinogenic effects may hinder SalA’s clinical usefulness [14,27,29,30]. SalA’s clinical usefulness is further impaired by bad pharmacokinetic properties with rapid gastrointestinal degradation, quick metabolic inactivation, and fast elimination [16,31,32,33,34,35,36]. At the oral route of administration, drug absorbance can only occur within the mouth mucosa by holding the SalA-containing product in the mouth for several minutes [16,36]. Vaporization or smoking of Salvia divinorum leaves or extracts with subsequent inhalation is a more efficient administration route and is often used by recreational users [16,17]. After bioabsorption, SalA quickly enters the central nervous system eliciting its clinical effects but leaving the CNS just as quickly [15,17]. SalA is metabolized by esterases, glucuronidases, and CYP enzymes [16,31,32,33] with fast elimination [34,35]. Especially blood esterases rapidly metabolize SalA via C2-ester cleavage to the main metabolite Salvinorin B (SalB) that is inactive at the KOR [15,16,35]. Of note sex differences in metabolism and elimination were observed with females showing slower metabolism and excretion [35]. As a result, SalA exhibits a short duration of effect [16,17,35], which likely contributes to SalA’s low toxicity but also hinders its usage as an analgesic drug.

To overcome SalA’s clinical drawbacks, a plethora of analogs were synthesized and tested [15,37], which led to the development of agonists [15,37,38,39,40], partial agonists [37,38,40,41] and antagonists [37,40,42,43] of the KOR and even to MOR modulators [37,44,45,46,47,48]. Despite a large number of around 600 available analogs the majority of these exhibit alterations at the C2 position or modifications of the furan ring [15] which resulted in fewer diverse ligand sets than desired. The rational design of new analogs is further hindered by the still unknown binding mode of SalA at the KOR which also hinders the understanding of the structure-activity relationship (SAR). Furthermore, most of the derivatives exhibit less affinity or activity than SalA itself [15]. The comparison of different analog series is hampered by the different assays used, the choice of different cell lines, and the reference ligand used. Due to the focus on KOR affinity and activity and the lack of in vivo experiments of analogs, it is hard to evaluate the safety profiles, hallucinogenic effects, addictive properties, and therefore the misuse potential of almost all analogs.

The rational design of new analogs to explore the chemical space around SalA is hampered by the still unknown binding mode of SalA at the KOR. Several different and conflicting binding modes for SalA at the KOR were proposed [22,25,49,50,51,52,53,54,55]. For example, Vardy and coworkers [49] as well as Roach and coworkers [50] postulated binding modes in which the furan moiety of SalA points upwards towards the extracellular portion of the KOR while Che and coworkers [22] and Kane and coworkers [25] postulated the opposed orientation. In contrast to the mostly vertical orientation of SalA within the KOR binding site, Vortherms and coworkers [53] predicted a horizontal orientation of SalA. The discrepancies within these results are partially caused by the lack of an active-state crystal structure of the KOR until 2018 [22]. SalA was then docked into KOR homology models [50,52,54] or active-like structures [49,51] derived from the inactive KOR crystal structure published in 2012 [20], which differs from the active-state crystal structure.

To solve the confusion about the SalA binding mode a comprehensive evaluation of SalA within the active-state KOR crystal structure with respect to available mutagenesis studies and SAR data is needed. In this paper, we carefully and rationally elucidate the binding mode of SalA to the active-state KOR taking into account available mutational and SAR data. We rationalize determinants for SalA receptor subtype selectivity as well as effects of ligand modifications for experimental measured affinity and activity at the KOR by using docking experiments, molecular dynamics simulations, and dynophore [56,57] analysis.

## 2. Results

### 2.1. Salvinorin A Binds above the Morphinan Binding Pocket of the Kappa Opioid Receptor

To elucidate the binding mode of Salvinorin A (SalA) at the kappa opioid receptor (KOR) we performed docking experiments of SalA to the KOR binding cavity of the prepared active-state KOR crystal structure (PDB-ID: 6B73 [22]). SalA binds above the morphinan binding site at the extracellular part of the KOR binding cavity with its C4-methyl ester interacting with the extracellular loop 2 (ECL2) while its furan moiety points downwards to the morphinan binding pocket (Figure 2). Unlike typical KOR ligands that always establish an ionic interaction with D138^3.32^ (superscript denotes Ballesteros–Weinstein numbering [19]) at the bottom of the morphinan binding site [20,21,22,23], SalA is solely stabilized in its position by hydrophobic contacts and hydrogen bonds (Table 1). The C4-methyl ester forms hydrogen bonds to E210^ECL2^, the C1-carbonyl moiety to Y312^7.35,^ and the C2-acetoxy group to Y313^7.36^. The furan moiety is stabilized via hydrophobic contacts with Y139^3.33^ and I294^6.55^, while C19 and C20 (both methyl groups) form lipophilic contacts with V118^2.63^, Y313^7.36^, or I316^7.39^, respectively. SalA sterically fits into the binding pocket shape with perfect complementarity.

Our proposed binding mode is in accordance with previous published mutational data: Several mutagenesis studies highlight the detrimental role of Y312^7.35^ and [25,49,54] and Y313^7.36^ [6,25,49] for SalA affinity and activity; in our model both residues anchor SalA in the KOR binding cavity. Participation of the ECL2 loop in SalA binding was proposed based on binding studies with chimeric opioid receptors [25], which agrees with C210^ECL2^ interacting with the C4 methyl ester of SalA via hydrogen bonding. V118^2.63^ and Y139^3.33^ were also proposed to affect SalA affinity [25,26,49] and stabilize SalA by hydrophobic contacts in the KOR binding site.

Despite the indisputable importance of mutagenesis studies for experimental validation of a proposed binding mode we need to carefully check whether the underlying studies were performed probe-dependently or probe-independently. Several studies found a strong decrease in the affinity and activity of SalA in a Y320^7.43^A KOR mutant [25,49,54] suggesting a SalA binding mode in which the ligand binds as deep in the orthosteric binding site as morphinan ligands do. Nonetheless, this phenomenon seems to be independent of the ligand tested albeit with strong differences in the chemical ligand space (SalA, nitrogen-containing small molecules, peptides) [49]. Thus, we surmise a ligand-independent conformational change of the KOR Y320^7.43^A mutant that led to the decreased affinities and activities experimentally measured. Indeed, Y320^7.43^ is in close proximity to W287^6.48^, a residue believed to play an important role in KOR activation as it directly interacts with F283^6.44^ of the conserved PIF-motif [22] and therefore the truncated side chain in the Y320^7.43^A mutant likely alters the orientation of W287^6.48^ and indirectly the KOR activation.

The Q115^2.60^A KOR mutant was found to strongly decrease SalA affinity and activity [25,49]. We did not observe interactions of SalA with Q115^2.60^ in our binding model but Q115^2.60^ is in close proximity to the C17-carbonyl of SalA. We, therefore, investigated the potential interaction of SalA with Q115^2.60^ through molecular dynamics simulations.

### 2.2. Molecular Dynamics Simulations Confirm Salvinorin A Binding Mode Obtained by Docking Experiments but Revealed Additional Interaction with Q115

To evaluate our SalA binding mode found in docking experiments and to assess whether Q115^2.60^ contributes to SalA binding in a dynamic investigation we performed molecular dynamics (MD) simulations of SalA bound to the KOR crystal structure and subsequent calculated dynamic pharmacophores (dynophores, Figure 3) [56,57]. Dynophores represent interactions dynamically. They consist of probability density point clouds representing the interactions detected over the course of MD simulations. MD simulations confirm our putative binding mode for SalA with an average ligand root mean square deviation (RMSD) of 2.9 Å over the five replica simulations (200 ns each). Detailed information about the RMSD of SalA and the KOR can be found in the Appendix A). SalA is stabilized within the binding pocket by highly frequent hydrophobic contacts and several hydrogen bonds (Table 2).

We observed a tendency for SalA to very slightly shift towards the TM2 over the course of the simulation and a tendency for Q115^2.60^ to reorient towards the C17-carbonyl of SalA facilitating hydrogen bonding (54.0%). This hydrogen bonding was not observed within the static docking results but agrees with previously published mutagenesis studies that showed decreased affinity and activity values for SalA with Q115^2.60^A KOR mutants [25,49].

All carbonyl groups of SalA participate in hydrogen bonding. Hydrogen bonding between the C4 methyl ester and C210^ECL2^ occurred in 94.1% of MD simulations. This most frequent hydrogen bonding interaction is likely important for SalA’s affinity as alterations in the C4-substituent mostly induce affinity and activity drops [15,58,59]. The C1-carbonyl group participates in highly frequent hydrogen bonding (79.0%) with Y312^7.35^. We detected weak hydrogen bonding between the C2-acetoxy group and KOR along MD simulations (17.6% to Y312^7.35^, Y313^7.36^). SalB, the main metabolite of SalA, exhibiting a C2-hydroxy group instead of a C2-acetoxy group is inactive at the KOR [15,16,35] rendering the C2-acetoxy group interaction of SalA important for SalA’s potency. Thus, the low frequency of the C2-acetoxy group interactions was unexpected. However, the absolute frequency of a particular interaction is less meaningful than the ratio of these interactions between analogs. Despite overall low occurrence, this interaction could still play a key role in the binding and activation of KOR by SalA. Off note, both residues that interact with the C2-substituent of SalA are non-conserved which likely contributes to the strong selectivity of SalA.

The hydrophobic contact between the furan moiety of SalA and L212^ECL2^ was not observed in static docking but agrees with the observation of poor binding of SalA to the L212^ECL2^A mutant [20].

The dynamic interaction analysis agrees with the results obtained by static docking with the addition of C17-carbonyl hydrogen bonding due to the reorientation of Q115^2.60^ and additional hydrophobic contacts to L212^ECL2^. Thus, the results are in accordance with mutagenesis studies too. Figure 3 shows the dynophore model (probability density point cloud representation of protein-ligand interactions) of the SalA-KOR complex.

### 2.3. Non-Conserved Residues Harboring Salvinorin A at the Kappa Opioid Receptor Lead to Receptor Subtype Selectivity of Salvinorin A

SalA is characterized by its strong KOR subtype selectivity without affinity towards the MOR, DOR, and the fourth OR called the nociceptin/orphanin FQ peptide (NOP) receptor [24,25,26]. Thus, SalA mediates its analgesic effect solely by KOR activation avoiding classical MOR side effects like respiratory depression, addiction, and constipation [5,7,8,9].

The three classical OR receptors (KOR, MOR, and DOR) share high overall sequence identity and similarity values that hamper subtype selectivity (Identity KOR/MOR = 56.3%; Identity KOR/DOR = 56.3%; Similarity KOR/MOR = 70.5; Similarity KOR/DOR = 68.2; all values for full sequence comparison, Appendix A). Especially the orthosteric binding site is highly conserved. This holds true for the GPCR class A family in general. On the contrary, the extracellular parts of these receptors are more diverse with several non-conserved residues. Thus, selective orthosteric ligands often additionally target extracellular regions of GPCRs to facilitate their subtype selectivity [60,61]. The recently discovered NOP is more distinct from the three classical ORs but still exhibits significant sequence identity and similarity values (Identity KOR/NOP = 48.9%; Similarity KOR/NOP = 62.6%; values for full sequence comparison). The less conserved structure leads to marked differences in the binding and activation profile of OR ligands binding to the NOP compared to binding to the classical ORs [62].

Docking experiments show that SalA binds above the morphinan binding site, the deepest part of the orthosteric binding site, within a less conserved site of the KOR. The residues V118^2.63^, I294^6.55^, Y312^7.35^, Y313^7.36^, and I316^7.39^ that participate in protein-ligand interactions with SalA in the KOR-SalA complex vary among the different OR subtypes and therefore likely influence SalA’s selectivity profile. While I316^7.39^ differs in the NOP but is conserved within the classical ORs the remaining before-mentioned residues vary within the classical ORs. Table 3 provides a comprehensive list of the non-conserved residues participating in SalA binding at the KOR.

In order to rationalize the determinants for SalA’s outstanding selectivity, we evaluated the impact of non-conserved residues within the SalA binding site for OR subtype selectivity. In order to account for the orientation of the non-conserved residues within the binding pocket, we superimposed the KOR-SalA complex according to its transmembrane region (OPM-Database entry: 6B73) with the active-state structures of MOR (PDB-ID: 5C1M [63]), DOR (PDB-ID: 6PT2 [64]) and NOP (Figure 4). Since there is no experimentally derived active-state NOP structure available, we used a homology model based on the active-state KOR crystal structure (PDB-ID: 6B73 [22]) already described in one of our previous publications [65].

V118^2.63^ at the top of KOR-TM2 forms hydrophobic contacts with C19 (methyl moiety between Rings A and B) of SalA. While nonpolar in the KOR, this region is polar in the remaining three receptors (MOR: N, DOR: K, NOP: D) and therefore would be unable to maintain this interaction with SalA in these receptors. The ionic character in DOR and NOP likely even pushes the lipophilic SalA scaffold away.

Y313^7.36^ establishes hydrogen bonding to SalA’s C2-acetoxy moiety in the KOR-SalA complex as well as hydrophobic contacts to C19 of SalA (methyl moiety between Ring A and B). The respective residues in the remaining ORs (MOR: H, DOR: H, NOP: R) are able to facilitate hydrogen bonding, but only if they are correctly oriented. The histidines in MOR and DOR are too far away to establish hydrogen bonds (MOR: 6.2 Å, DOR: 6.4 Å; distance measured between NE2 of histidines and C2-acetoxy carbonyl oxygen atom of SalA) and also to form hydrophobic contacts (MOR: 6.5 Å, DOR: 6.9 Å; distance measured between CE1 of histidines and C19 of SalA). They therefore cannot facilitate the interactions possible at KOR. In our NOP homology model, the arginine was predicted to point away from the ligand binding site towards E295^7.29^ at the top of TM7. Thus, it likely would not contribute to SalA binding towards the NOP.

The C1-carbonyl moiety of SalA forms hydrophobic contacts to Y312^7.35^. From the three remaining ORs (MOR: W, DOR: L, NOP: L) only tryptophan in the MOR could form hydrogen bonds if positioned correctly within the binding pocket. The receptor superimposition revealed that the tryptophan in MOR is too distant for hydrogen bonding analog in the KOR-SalA complex (5.2 Å between NE1 of W320^7.35^ and C1-carbonyl oxygen atom). Thus, none of the ORs can mimic the interactions of Y312^7.35^ to SalA in KOR.

The furan moiety of SalA is stabilized by hydrophobic contact with the conserved Y139^3.33^ and the non-conserved I294^6.55^. The respective residues for I294^6.55^ in the remaining receptors (V for all three receptors) are all hydrophobic as well and only differ from I294^6.55^ in the truncation of one methyl group. Within the receptor superimposition, all residues are within the range of 5.9 Å for hydrophobic contacts set as the default range in Ligandscout 4.4.3 (Inte:Ligand, Vienna, Austria) [66,67] albeit scarce (MOR: 5.4 Å, DOR: 5.9 Å, NOP: 4.6 Å, measured between C15 of SalA and CG2 of valine in MOR and DOR and CG1 of valine in NOP). We surmise that I294^6.55^ hardly contributes to SalA selectivity. The effect of the beforementioned residues is more pronounced.

The position 7.39 participates in hydrophobic contacts with C20 (methyl moiety between Ring B and C) at the KOR-SalA complex and is conserved within the classical ORs (KOR: I316^7.39^, MOR: I324^7.39^, DOR: I304^7.39^) and only differs in the NOP structure (T305^7.39^). The isoleucine is positioned similarly in all three classical ORs with all residues capable to facilitate hydrophobic contacts. Despite being rather polar the side chain methyl group of the threonine in the NOP homology model is oriented in that it can participate in hydrophobic contacts, albeit scarce again (5.9 Å betweenCG2 of T305 and C20 of SalA). The non-conserved residue 7.39 therefore likely has a minor role in SalA KOR selectivity.

After careful evaluation of non-conserved residues participating in SalA binding at the KOR, we surmise positions 2.63, 7.35, and 7.36 are responsible for SalA selectivity as the interactions within the KOR complex cannot be mimicked by the remaining receptors.

To date, the active-state MOR crystal structure (PDB-ID: 5C1M [63]) is the only experimental solved OR structure with longer parts of the flexible extracellular N-terminus solved. In the superimposition of the KOR-SalA complex and MOR, a clash of SalA (Ring A, C2-acetoxy group, C4-methyl ester) and the N-terminus of MOR is observed. As the MOR structure was derived with a morphinan-based orthosteric ligand binding deeper in the receptor and due to the flexible character of the N-terminus the consequence of this potential clash cannot be fully evaluated. Furthermore, as the N-termini of KOR, DOR and NOP are not solved and therefore their respective positions are unknown the impact of the N-terminus position of the binding and selectivity profile of SalA cannot be estimated.

### 2.4. Salvinorin A Binding Mode Is in Agreement with Previous Published Structure-Activity-Relationship Data

Next, we evaluated our binding hypothesis in the context of available structure-activity relationship (SAR) data. Due to around 600 experimental tested analogs [15] of SalA (**1**) being available, sufficient data is available to evaluate and further assess our model. Thus, we carefully selected several analogs of SalA to explain the general effects of specific substitution patterns on the affinity and activity of SalA analogs. Most SalA alterations lead to strong affinity and activity drops and only a little number of analogs with improved properties are known [15,39,68,69,70,71]. Figure 5 shows the structures of all ligands discussed in this section while Table 4 lists the respective experimental data.

#### 2.4.1. C2-Analogs of SalA

The majority of available SalA analogs host alterations of the C2-acetoxy group of SalA [15]. Several ester analogs including ethers, amides, amines, and carbamates are known and others are available [15,38,68,70,73].

The common metabolic inactivation of SalA represents the C2-acetoxy group cleavage leading to SalB (**2**) with a C2-bound hydroxy group [16]. Despite the wide acceptance of SalB as the inactive metabolite of SalA conflicting experimental data were measured with K_i_ values between 111 nM and >10,000 nM [15,38,47,68,71,72,73] and EC_50_ values between 2.4 nM and 492 nM [71,78]. However, compared to SalA, SalB binds to the KOR in a much weaker way. Docking of SalB into the KOR binding cavity reveals an almost identical binding mode of SalB compared to SalA (Steric overlap of 87%) with almost identical interactions, albeit the lack of any C2-hydroxy group interactions (Appendix A). This finding renders the C2-acetoxy hydrogen bonding an important requirement for KOR affinity and activity of SalA despite its moderate frequency in MD simulations (17.6%).

Several authors postulate a size restriction hypothesis for the C2-substituent with spacious lipophilic groups resulting in affinity drop [14,41,79]. We replaced the acetoxy group at C2 of SalA with a more spacious pivaloxy moiety (**3**) [72] or a cyclopropanecarboxylic moiety (**4**) [71,72] as both analogs were measured to have an abolished affinity towards KOR [15,71,72] and docked both analogs into the KOR. Docking experiments show reduced interactions for both analogs with **4** lacking both hydrogen bonds at the C1-carbonyl moiety and at the carbonyl oxygen of the C2-cyclopropanecarboxylic moiety while its cyclopropyl moiety does not form any hydrophobic contacts (Appendix A). The pivaloxy-analog **3** is capable of stabilizing its C2-rest in hydrophobic contacts with I294^6.55^, but lacks the C1- and C2-hydrogen bonds as well (Appendix A). The C1-carbonyl hydrogen bonding occurred in SalA MD simulations with high frequency (79.0%) and therefore likely contributes to SalA’s experimental activity. Its lack of both analogs likely diminishes affinity. The hydrogen bond at the C2-acetoxy group of SalA was detected with lower frequency in SalA MD simulations (17.6%), but due to its important role in SalA metabolic inactivation, it is considered important as well. The combined lack of both interactions likely rationalizes the completely abolished affinity of the two analogs. According to the docking poses of the analogs, the C2-substituents are positioned in the middle of the central binding cavity pointing towards TM6. Within this region, there are only a few lipophilic residues present (mainly I294^6.55^) but several charged residues (E297^6.58^, K227^5.39^, E209^ECL2^). Additionally, the C2-substituents would be surrounded by water filling the empty parts of the binding cavity. Together, this explains the experimentally measured loss of affinity by substituting the C2-acetoxy group of SalA with bulkier lipophilic moieties as they hardly can form interactions resulting in enthalpically unfavorable binding poses. A size restriction rule due to a small pocket that only can accommodate a limited number of atoms cannot be applied.

An outstanding feature of SalA is its nitrogenless structure, but several studies tested the effect of nitrogen introduction at different positions of SalA [15,38,39,41,58,73,80,81,82]. At C2 the introduction of a positively charged nitrogen often led to strong decreases in KOR affinity and activity but some analogs tolerate the introduction of the positive charge [15,38,39,73,80,81]. This finding agrees with the C2-acetoxy group of SalA being oriented towards a region with multiple negatively charged ligands (E297^6.58^, E209^ECL2^) that could interact with positive amines.

Beguin and co-workers [73] tested a series of alkyl amines at the C2-position with **5** being the best-tolerated amine (K_i_ 17.6 nM, about 14-fold diminished affinity compared to SalA) with an isopropylamine moiety at C2. Compared to the binding mode of SalA, **5** is shifted towards TM5 establishing charge interactions with E297^6.58^ and E209^ECL2^, but loses hydrogen bonding at the C4 position (Appendix A). The isopropyl moiety at the amine is likewise surrounded by the negative residues and does not take place in any hydrophobic contacts. The lack of the hydrogen bond at C4 combined with the overall shift within the binding pocket likely caused the decrease in affinity, but the ionic interactions stabilizing **5** from two sides seem to rescue its affinity towards KOR.

Although the alteration of the SalA structure mostly leads to impaired affinity and activity values measured in experiments, several examples of analogs with improved properties are known. One of such improved analogs is **6** with an ethoxymethyl ether at C2 [15,69,70] being the most affine ligand of a series of alkoxymethyl ether derivates tested by Munro and coworkers [69]. Docking of **6** to KOR revealed an extended interaction pattern when compared to the KOR-SalA complex (Appendix A). The oxygen atoms of the ethoxymethyl ether establish hydrogen bonding to Y312^7.35^ and Y313^7.36^ and the ethyl group is stabilized by hydrophobic contacts to Y312^7.35^ and L309^7.32^. Hydrogen bonding to Q115^2.60^ that was not seen in the static investigation of SalA but in MD simulations (79.0% frequency) occurred also in the KOR-**6** complex (to C1-carbonyl moiety of **6**). The new interactions while maintaining the SalA interaction pattern rationalizing the increased affinity values measured for **6**.

22-thiocyanato-SalA, also called RB-64 (**7**, RB-64), is one of the most prominent SalA analogs as it shows G protein bias and therefore represents a potential analgesic compound with an improved safety profile [6,83]. RB-64 (**7**) was originally designed by Yan and coworkers [55] as a covalent KOR agonist, but the postulated covalent binding mode with C315^7.38^ is poorly supported by experimental data. The experimental data for the wash-resistant readout in the binding assay is not shown within the publication, the experiments were conducted under unphysiological conditions with a huge amount of ligand (10 h and 20 μM, 4 °C), and the postulated covalent bond between RB-64 and C315^7.38^ is quite large (4.9 Å). However, White and coworkers postulate a non-covalent binding mechanism for RB-64 administered in vivo [83]. The experimentally measured affinity of RB-64 towards KOR differs in a probe-dependent manner. RB-64 (**7**) shows improved affinity compared to SalA in [^3^H]U69,593 competition assay [55,68] but slightly impaired affinity in [^3^H]diprenorphine competition assay [55].

We performed non-covalent docking of RB-64 into the active state KOR crystal structure as we doubt the covalent binding mode proposed by Yan and coworkers [55] and due to the lack of a free cysteine residue around the C2-acetoxy group of SalA. Our non-covalent docking experiments reveal a highly similar binding mode of RB-64 compared to SalA, but with lacking interactions at the C2-substituent (Appendix A). The absence of C2-substituent interactions is questionable as RB-64 shows improved affinity compared to SalA in certain experiments. We, therefore, performed 1 µs (five replicas à 200 ns) of MD simulations for the RB-64-KOR complex to investigate the interactions in a dynamic fashion. The MD simulations reveal highly similar interactions between the RB-64-KOR complex and the SalA-KOR complex, but with the exception of slightly increased C2-substituent interactions in the case of RB-64 (Table 5). The thiocyanate of the C2-substituent only weakly participates in hydrogen bonding (8.6%) but probably serves as an anchor to stabilize the position of the carbonyl moiety of C2. Compared to SalA the interaction frequency of the C2-carbonyl moiety is increased by around 5% to 22.2% in total. As mentioned earlier, the absolute frequency is less meaningful than the frequency shift between close derivatives in order to explain affinity differences obtained in experiments. Therefore, the small increase in the interaction frequency observed in MD simulations can explain the improved affinity of RB-64 in certain experiments. Information about the RMSD von RB-64 and the protein-heavy atoms can be found in Appendix A.

We further performed several relative binding free energy (RBFE) calculations based on the docking poses of SalA and RB-64 (FEP in Maestro [84,85], openfe [86]) and absolute binding free energy calculations of both complexes (Yank [87]). All methods predicted the binding of RB-64 at KOR more favorable than the binding of SalA to KOR. More information about the energy calculations can be found in the Appendix A and the Section 4.

#### 2.4.2. C4-Analogs of SalA

In contrast to C2, charges at C4 are not tolerated in SalA and lead to drastically reduced affinity values [15,58,59,73,74]. The C4-substituent of SalA is positioned near ECL1 and ECL2 without any charged residues in reach and with a space restriction that only allows growing to the extracellular side. Thus, the introduction of a charge at C4 likely introduces an unfavorable negative repulsion that forces the ligand to adopt a new orientation resulting in reduced affinity compared to SalA. We docked one analog with a negative charge at C4 (carboxylate moiety, **8** [58,59,74]) and one with a positive charge (methylaminomethyl moiety, **9** [73]) both without any measurable affinity at 10 µM concentration to the KOR. As expected, we found no reasonable docking poses for both compounds. In order to fulfill charge interactions, both compounds adopt orientations completely dissimilar from SalA with scaffold reorientations (Appendix A). The absence of a rational binding pose explains the lack of experimentally measured affinities for compounds containing a charge at C4.

Lee and coworkers [58] tested a series of alkyl esters at the C4 position and found a complete loss of affinity by any extension of the methyl group. Even the addition of one methyl group (ethyl ester) led to an absence of any affinity. Thus, we investigated a possible size exclusion effect of the C4 moiety. Docking of the ethyl ester (**10**) [58] as the smallest alkyl ester with abolished affinity and the bulkier pentyl ester (**11**) [58] cause ligand shifts within the KOR binding site towards TM5 to accommodate the C4-substituent within a more lipophilic pocket at TM2/3 (T111^2.56^, V118^2.63^, W124^23.50^, V134^3.28^, I135^3.29^) instead of the hydrophilic environment of ECL1/ECL2, where the methyl ester of SalA is positioned (Appendix A). The ligand shift towards TM5 causes the loss of all hydrogen bonds in the case of the ethyl ester, while the pentyl ester can at least rescue hydrogen bonding at the C1-carbonyl position with Y312^7.35^. The strongly diminished interaction pattern with loss of C2- and C4-hydrogen bonding in both cases and the additional C1-hydrogen bonding loss in case of **10** rationalizes the observed affinity lack for all alkyl esters except the methyl ester of SalA.

#### 2.4.3. C12-Analogs of SalA (Furan-Analogs)

The furan moiety at C12 of SalA is positioned the deepest in the KOR binding cavity and is stabilized by hydrophobic contacts with Y139^3.33^ and I294^6.55^. Simpson and coworkers [42] found that the omission of the C12-substituent (**12**) results in a 1789-fold decrease in binding affinity. At the same time, the replacement of the furan by a carboxylic moiety (**13**) only causes a small affinity drop (22-fold) [41]. Thus, we surmise the furan to be an anchor point for the ligand with establishing favorable but not necessarily hydrophobic contacts.

To rationalize how SalA tolerates the replacement of the furan by a carboxylic moiety we docked **13** into the KOR binding site (Appendix A). **13** takes place in extensive protein-ligand interactions with establishing hydrogen bonds to six interaction partners (C210^ECL2^, S211^ECL2^, L212^ECL2^, K200^ECL2^, K227^5.39^, Y312^7.35^) and ionic interactions with two residues (K200^ECL2^, K227^5.39^). The C4-methyl ester does not participate in the interaction pattern. The carboxylate shifted towards TM5 and is positioned more extracellularly than the furan of SalA. The furan serves as an anchor for the remaining ligand scaffold without establishing crucial interactions. Thus, the C12-substituent shift is likely tolerated and the centration between the two lysines anchors the remaining ligand scaffold of **13** in a conformation with hydrogen bonding to almost all residues also attacked by SalA with only interactions to Y313^7.36^ missing. Keeping crucial interactions from SalA with additional hydrogen bonding and anchoring ionic interactions illuminates the moderate toleration of a carboxylate in a C12 position. Of note, **5** (isopropylamine at C2) is anchored between two opposing charged residues (E297^6.58^, E209^ECL2^) with no C4-interactions and moderately tolerated as well (14-fold diminished affinity compared to SalA). Thus, the deficit of the C4-substituent interactions seems to be less severe in cases where the C1- and C2-substituent interactions are maintained.

There is no clear correlation between the size or polarity of C12-alterations and affinity. Besides the small polar carboxylate substituent at C12 in **13** being tolerated the bulky m-carboxamidobenzoyl moiety in **14** [75] is tolerated as well with only a 5-fold affinity drop. The bromination of C16 in **15** [41,42,88] at the furan ring increasing steric size and hydrophobicity does not influence binding to KOR at all while the replacement by bulky greasy naphthalene in **16** [76] diminishes affinity completely. Docking of **14** and **15** to the KOR revealed almost identical binding poses and the maintenance of all interactions was also detected in the KOR-SalA complex but with additional interactions respectively (Appendix A). The m-carboxamidobenzoyl moiety of **14** participates in hydrogen bonding to D138^3.32^ with its carboxamide substructure and to Y139^3.33^ with its benzoyl substructure while the bromine atom of **15** established hydrophobic contacts to Y139^3.33^. The almost identical binding modes of these two analogs with the one of SalA rationalizing the toleration of both substitution patterns according to KOR affinity. In contrast, docking of **16** resulted in the loss of all hydrogen bonds except the interaction of the C1-carbonyl moiety with Y313^7.36^ while the naphthalene moiety is shifted towards TM5 (Appendix A). These docking results illustrate the difficulty of rationally designing newly tailored SalA-analogs without prior knowledge about the actual binding mode of SalA within the KOR by only taking physicochemical properties into account. The combination of favorable physicochemical properties and the right orientation within the binding pocket facilitates improved affinity while only one requirement satisfied can still result in no binding at all.

#### 2.4.4. SalA Derivatives with Modified Scaffolds

Most alterations of the SalA structure focus on the adaption of the C2-, C4, and C12-substituents [15,37]. Nonetheless, some SalA-analogs with alterations within the SalA-ring-system are known including reduction of the carbonyl moieties at positions 1 and 17 as well as ring-opening [15,42,43,71,77,78,81,89].

The complete reduction of the C17-carbonyl moiety in **17** has almost no effect on KOR affinity [77] rendering the C17-carbonyl interactions favorable but dispensable. Docking of **17** into the KOR shows an almost identical binding mode compared to SalA with the same interactions as detected in the KOR-SalA complex and thus explains the almost unaltered biological data (Appendix A).

The effect of omitting moieties at C1 is contradictory. While the complete omission of substituents (**18**) only shows a small impact on KOR affinity and activity [43,77], the change of the carbonyl moiety to a hydroxyl group (**19**) leads to a 281-fold affinity drop [43,77]. When docking **18** and **19** into the KOR we observed similar docking poses compared to SalA, but with slight differences (Appendix A). **18** forms the same protein-ligand interactions as SalA with only the C1 interaction missing, while **19** shows altered interactions. **19** lacks the C2-interaction to Y313^7.36^ as its C2-acetoxy group is differently oriented and establishes a hydrogen bond to Y312^7.35^ with its C1-hydroxyl group instead. The fact that the complete loss of the C2-substituent abolishes KOR activity completely [78] rationalizes the strong affinity drop for **19**. As **18** mainly differs in the interaction pattern for C1 and was shown less critical for affinity [43,77] it can rescue its affinity in experiments.

## 3. Discussion

In this study, we presented a putative binding mode for SalA bound to the KOR active-state crystal structure (PDB-ID: 6B73 [22]). To the best of our knowledge, this is the first paper where the binding mode of SalA is systematically modeled within the active-state KOR crystal structure available since 2018. We comprehensively evaluated the structure-activity relationship and mutational data available to generate our model. We further elucidated our binding hypothesis by extensive molecular dynamics simulations and identified relevant structural OR subtype selectivity determinants.

In previous publications, a variety of different binding modes for SalA (and close derivatives) bound to the KOR were postulated but with conflicting orientations [20,22,25,49,50,51,52,53,54,55] causing confusion within the community. The binding poses were guided by mutational data available for SalA at the KOR rather than SAR data. Most studies [22,25,49,50,51,52] predicted a vertical orientation of SalA within the KOR binding pocket to account for the far distance between the residues highlighted as important for SalA’s affinity and potency at KOR in mutagenesis studies (from Y320^7.43^ at the mid of TM7 to residues belonging to ECL2). However, horizontal binding modes were proposed as well [53,54]. The majority of studies evaluating SalA’s binding mode were conducted before the publication of the active-state KOR structure (PDB-ID: 6B73 [22]) in 2018. Homology models based on other active-state GPCR structures (like rhodopsin [52] or MOR [50]) or active-like KOR models [20,49,51] based on the inactive-state KOR crystal structure (PDB-ID: 4DJH [20]) were used instead. Those KOR models differ in overall receptor architecture as well as side-chain orientations from the experimentally solved active state X-ray KOR crystal structure used by Che and coworkers [22] and within this study. These differences in the KOR model are used to contribute to the different binding modes proposed for SalA. Yan and coworkers [54] for example generated a KOR model based on the inactive-state rhodopsin crystal structure with Y119^2.64^ pointing towards the binding site while Y312^7.35^ and Y313^7.36^ are oriented more outward ending up with a rather horizontal SalA binding mode to agree with mutagenesis studies. In the active-state crystal structure, the orientations of Y119^2.64^, Y312^7.35^, and Y313^7.36^ are opposed rendering a horizontal binding mode of SalA implausible.

Results of mutagenesis studies should be treated with caution. The effects can be divided into probe-dependent and probe-independent or direct and indirect. Mutations can affect ligand binding and receptor activation by direct interactions with the ligand but also due to alterations in conformations of neighboring residues. Y320^7.43^ and Y119^2.64^ were both highlighted as important for SalA affinity/activity based on mutagenesis studies [25,49,54]. Y119^2.64^ is oriented outward the binding cavity in the active-state X-ray KOR crystal structure which renders direct protein-ligand interactions doubtful. The strong decrease in KOR activity by Y320^7.43^A mutation occurs not only for the nitrogen lacking SalA but also for chemically distinct nitrogen-containing ligands and opioid peptides turning the effect likely probe-independent. The impact of both mutations consists probably of influencing neighboring residues, particularly Y313^7.36^ in the case of Y119^2.64^ and W287^6.48^ which in turn influences the conserved PIF motif in the case of Y320^7.43^. Thus, the evaluation of a binding mode solely by mutational data is less valuable than taking SAR data additionally into account.

Several studies postulate a binding mode in which the furan of SalA points towards the extracellular side of KOR [49,50,51], which is in contrast to our binding mode and those of several other publications [22,25,52]. Roach and coworkers [50] for example used docking of SalA into the active-state KOR homology model based on the active-state MOR crystal structure (PDB-ID: 5C1M [63]) resulting in a binding mode in which the furan points upwards the receptor cavity while the C2-acetoxy and C4-methyl ester of SalA is positioned deep in the receptor orthosteric binding site. The C4-methyl ester moiety of SalA points towards the negatively charged D138^3.32^. However, experimental testing of SalA analogs revealed the intolerance of positively charged amino groups at the C4 position [15,58,59,73,74] rendering the close proximity of the C4 group to D138^3.32^ implausible. Vardy and coworkers [49] used flexible docking into the inactive-state KOR crystal structure (PDB-ID: 4DJH [20]) and obtained a similar binding mode as Roach and coworkers with the furan pointing upwards the binding cavity but with a permutation of the C2-acetoxy and the C4-methyl ester groups so that the C2-moiety directly interacts with D138^3.32^. This binding mode is supported by the fact that positively charged amino substitutions at the C2 position could be tolerated [73]. However, the D138^3.32^A mutation does not affect the binding of SalA to the KOR [25]. Additionally, cystein-substitution analysis and CoMFA analysis gave hints to the proximity of C2-substituents to E297^6.58^ [52,54] which agrees with our postulated binding mode but conflicts with any binding modes having the furan oriented extracellular and the C2-acetoxy group intracellular.

In 2018 Che and coworkers [22] published the active-state KOR X-ray crystal structure and postulated a binding mode for SalA, albeit without any validation. The published binding mode is similar to our proposed binding mode. In their model, SalA is positioned vertically in the KOR binding cavity with its furan moiety pointing intracellular interacting with Y139^3.33^ while the C1-carbonyl moiety, the C2-acetoxy moiety, and the C4-methyl ester group point upwards interacting with Y312^7.35^ (C1, C2) and C210^ECL2^ (C4). Despite the high similarity between their binding mode and the binding mode of SalA described within this study Che and coworkers did not provide any validation of their putative binding hypothesis. Thus, the confusion within the community about the correct orientation of SalA in the KOR binding site still hold on and kept the need for this study up.

Of note, even after the elucidation of the SalA binding mode, the rational design of SalA analogs with improved clinical usefulness as analgesics is hampered by the lack of data according to the side effects of SalA analogs, first of all, their hallucinogenic properties, but also the common KOR related side effects like anhedonia, sedation, and locomotor incoordination. The side effect profile of SalA analogs needs to receive more attention and should be considered in further studies. G-protein-biased KOR agonists are promising safer analgesics as they likely sow an improved safety profile [6]. Some G-protein-biased SalA analogs are already known, including RB-64 (22-Thiocyanatosalvinorin A), and further attempts at understanding and utilization of G-protein bias may occur in SalA analogs with desired clinical properties.

## 4. Materials and Methods

### 4.1. Protein Preparation

All used receptor structures within this study were originally retrieved from the Protein data bank (PDB) [90] and subsequently processed using MOE 2020.0901 [91]. The respective PDB entries for the active-state X-ray crystal structures are 6B73 for KOR, 5C1M for MOR, and 6PT2 for DOR [22,63,64]. In the case of 6B73 (KOR) and 6PT2 (DOR) which were solved as dimeric receptors only the monomeric chain with the better resolution was processed while the remaining one was deleted. All fusion proteins (KOR: nanobody, MOR: antibody fragment, DOR: thermostabilized cytochromeb562 (BRIL)), cocrystallized lipids, and solvent molecules were deleted. To restore the human wild-type receptor sequence the receptor sequences were remutated according to the respective UniProt-Databank entries [92] (UniProt-IDs: P41145 for KOR, P35372 for MOR, and P41143 for DOR). The loop modeler implemented in MOE was conducted to model partially missing loop structures (ECL2, ECL3, ICL3) of KOR and missing side chains of all receptor structures. Improved geometric properties of the receptor structures were facilitated by careful energy minimization of atom clashes and Ramachandran outliers [93] using the OPLS-AA force field [94]. Finally, the receptor structures were protonated at 300 K and a pH of 7 using the Protonate3D application [95] implemented in MOE.

### 4.2. Docking

The structure of Salvinorin A (SalA) was retrieved from the PubChem database [96] and protonated using the Protonate3D application [95] within MOE. Docking experiments of SalA into the receptor structures were performed by conducting GOLD v5.2 [97]. The binding site was defined by a 30Å sphere around D3.32-γC (KOR: D138, MOR: D149, DOR: D128) encompassing the hole extracellular half of the receptors. 30 docking runs per experiment were performed resulting in 30 diverse solutions. ‘Diverse solution’ means that the calculated poses exhibit a root mean square deviation of at least 1.5 Å. The docking solutions were scored according to the GoldScore docking function [98,99]. The search efficiency was set to 200% and pyramidal nitrogen atoms were allowed to flip during the calculations. For docking of SalA into the KOR structure, a distance constraint between C20 of SalA and the phenolic oxygen atom of Y313^7.36^ (maximum 4 Å) was set to ensure binding of SalA around the putative binding site derived by mutagenesis studies.

The generated docking poses were subsequently energy minimized within their protein environment using the MMFF94 force field implemented in LigandScout [66,67] and visually inspected. The final binding hypothesis of SalA was selected according to the geometric properties of SalA within the binding pocket, the number of interactions, and according to information derived from mutagenesis studies as well as experimental testing from SalA analog series. In particular, the residues Q115^2.60^, V118^2.63^, Y119^2.64^, Y139^3.33^, Y312^7.35^, and Y313^7.36^ as well as residues from the ECL2 were considered as putative binding site residues [25,26,49,54] and the C4-methyl ester group must not point towards TM5/TM6 region, i.e., towards several charged residues as neither positive nor negative substituents at the C4-position are tolerated [15,58,59,73,74]. The putative binding poses of SalA analogs were selected according to the Gaussian shape similarity score [100,101] towards SalA measured in LigandScout and the number of interactive features.

### 4.3. Molecular Dynamics Simulations and Generation of Dynophores

Molecular dynamics (MD) simulations of SalA and RB-64 were prepared with Maestro v2020 [102] and performed with Desmond v2020-4 [85] in five replicates of 200 ns each. The Ligand-KOR complex was put in a rectangular simulation box that spans 10 Å to each receptor side and subsequently embedded in a POPC (1-palmitoyl-2-oleoylphosphatidylcholine) membrane according to the OPM database [103] entry for the active KOR structure (PDB-ID: 6B73). The remaining space within the simulation box was filled up with TIP3P water molecules [104] and ions (Na^+^, CL^−^) in an isotonic mixture (0.15 M). For system parametrization, the Charmm36 force field [105] was implemented into Maestro-setup using viparr-ffpublic [106,107]. The simulation run under NPT ensemble conditions, i.e., with a constant number of particles, constant pressure (1.01325 bar), and constant temperature (300 K).

For subsequent simulation analysis, the protein was centered and the trajectory (1000 frames) was aligned onto the backbone heavy atoms of the first sampled protein conformation using VMD v1.9.3 [108]. Additional to the visual inspection of the MD simulations dynamic pharmacophores (dynophores) were calculated with the in-house developed Dynophore tool [56,57]. Dynophores can be visualized within LigandScout as the dynophore algorithm is implemented in the same ilib framework as LigandScout. Only interactions occurring in at least 5% of the simulation time were considered for evaluation.

### 4.4. Energy Calculations

We performed several different energy calculations implemented in different software applications for calculating the relative binding free energy (RBFE) and absolute binding free energy (ABFE) of SalA and Rb-64 bounded to KOR. We used the docking poses for calculations.

Openfe [86] is an open-source python package (https://github.com/OpenFreeEnergy/openfe, accessed on 1 December 2022) that uses alchemical transformation together with replica exchange simulations to predict RBFE values. The software utilizes OpenMM [109] for conducting the simulation of 5 ns with NPT conditions and additional 2 ns equilibration. The protein was embedded in the TIP3P water model [104,110] with 0.15 M NaCL. The default ‘RelativeLigandTransformSettings’ were used encompassing 11 λ windows, 1 bar pressure, and 298.15 Kelvin as temperature among others. For the protein the amber99sb forcefield [111] and for the ligand, the openff-2.0.0 forcefield [112] was applied.

The Ligand FEP panel only implemented in the academic version of Desmond, obtained from D. E. Shaw Research [84,85], uses free energy perturbation calculations for the prediction of the RBFE. 11 replicates of 11 λ-intervals each are performed. The simulations of 5 ns each with an additional equilibration time beforehand are conducted under NPT conditions and the SPC water model [113]. The OPLS2005 forcefield [114,115] was applied.

The open-source python package YANK [87] (https://github.com/choderalab/yank, accessed on 1 December 2022) predicts absolute binding free energy values by conducting alchemical free energy calculations. The whole thermodynamic cycle with its solvent phase (ligand and receptor separated and solvated where the ligand is coupled or decoupled) and the complex phase (ligand bound to the receptor where ligand interacts with protein or is decoupled but restrained in a harmonic manner not to move too far from the binding site) are calculated. We performed calculations with explicit water (TIP4P-EW water model [116]). For accurate long-range treatment of electrostatic interactions particle mesh ewald (PME) method [117,118] was applied with a cutoff at 9 Å. For simulations, the ff14SB forcefield [119] for the protein and the gaff2 forcefield [120] for the ligand were applied.

## 5. Conclusions

In this study we present the binding mode of SalA to the active-state KOR crystal structure, discuss and present this carefully elucidated binding mode using available mutational and structure-activity data of SalA derivates, and investigate SalA binding using extensive molecular dynamics simulations. The consistency of the binding conformations led to the identification of selectivity determinants towards the other OR subtypes.

SalA adopts a vertically binding mode within the KOR binding site with its furan moiety pointing towards the receptor center while its C2- and C4-substituents span towards the extracellular site. The SalA-KOR complex is stabilized by extensive hydrogen bonding and hydrophobic contacts. Based on this binding mode we rationalize the affinity and activity data of several analogs of SalA. We surmise the C2-substituent interactions as important for SalA and its analogs to be experimentally active, albeit with moderate frequency within MD simulations of SalA. Protein-ligand-interactions at the C17-carbonyl group are rendered less important than those involving C1-, C2-, and C4-substituents as the complete reduction of the former moiety does not alter the affinity of the analog compared to SalA. The furan moiety likely serves as an anchor point for the ligand as modification of this group mostly decreases affinity but the replacement by a negatively charged carboxylate group is somehow tolerated.

SalA interacts with several non-conserved residues in the KOR binding site. We surmise the non-conserved residues 2.63, 7.35, and 7.36 are responsible for SalA’s excellent OR subtype selectivity with no binding to MOR, DOR, and NOP. The respective residues in the MOR, DOR, and NOP are unlikely to maintain the interactions observed at the SalA-KOR complex.

We discussed our proposed binding mode of SalA in light of former publications that postulate a majority of different binding hypotheses for SalA at the KOR. We highlighted the difficulty of choosing the right KOR model for subsequent docking studies, the necessity to consider possible probe-independent effects within mutational studies, and the need for SAR data to be considered to determine a binding mode in silico. We show that SalA needs to adopt a vertical binding mode with the C2- and C4-moieties pointing toward the extracellular side instead of the central core of KOR. We are confident that the new insights in the binding mode of SalA at the KOR will facilitate rational design of new analogs with improved properties and therefore promote the development of clinically useful analgesics based on SalA.

## Figures and Tables

**Figure 1 molecules-28-00718-f001:**
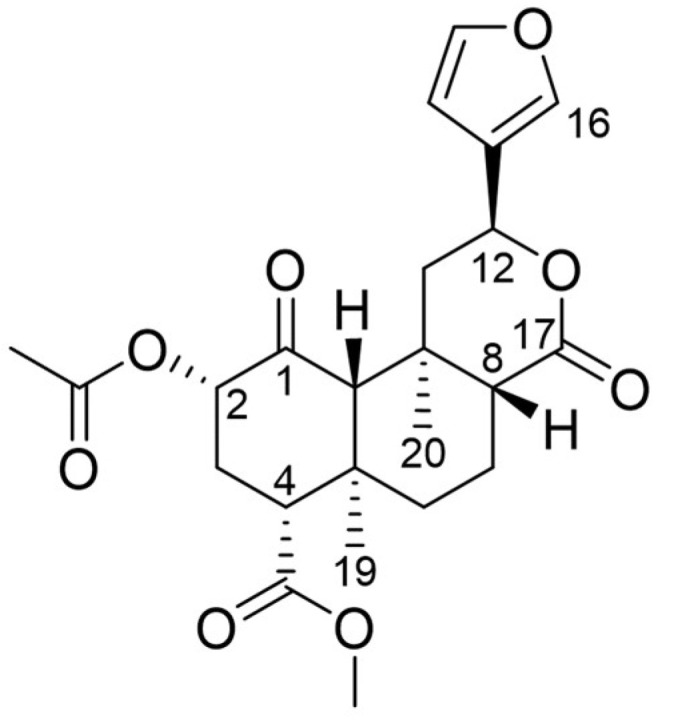
Chemical structure of Salvinorin A (SalA).

**Figure 2 molecules-28-00718-f002:**
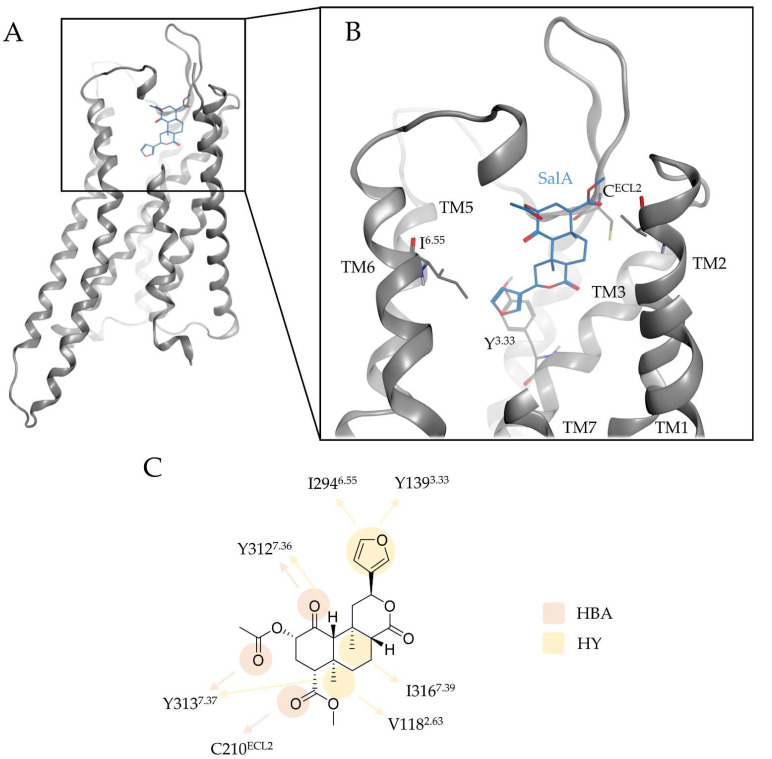
Binding mode of SalA at the KOR. (**A**) shows the overall architecture of KOR bound to SalA while (**B**) highlights the binding pocket of SalA at the KOR. (**C**) depicts the protein-ligand interactions between SalA and KOR. Y^3.33^ denotes to Y139^3.33^, C^ECL2^ to C210^ECL2^, and I^6.55^ to I294^6.55^. TM denotes to transmembrane helices, HBA to hydrogen bond acceptor, and HY to hydrophobic contact.

**Figure 3 molecules-28-00718-f003:**
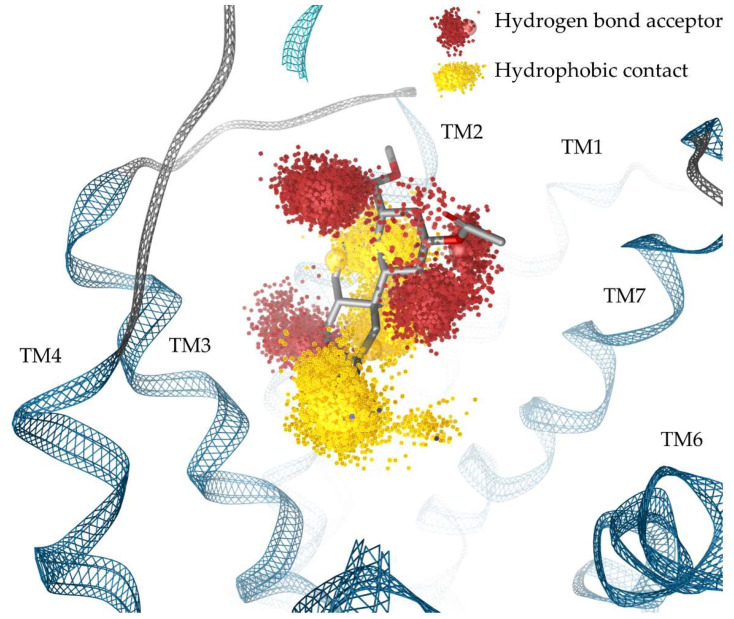
Dynamic pharmacophores (dynophores) of protein-ligand interactions between SalA and KOR (probability density point cloud representation). TM denotes to transmembrane helices.

**Figure 4 molecules-28-00718-f004:**
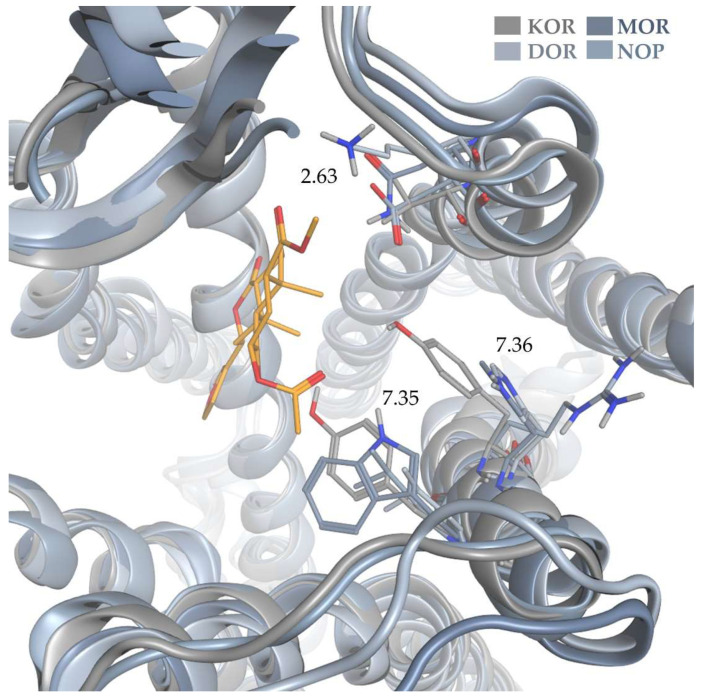
Superimposition of SalA bound to the KOR and the active-state crystal structures of MOR and DOR as well as the active-state homology model of NOP.

**Figure 5 molecules-28-00718-f005:**
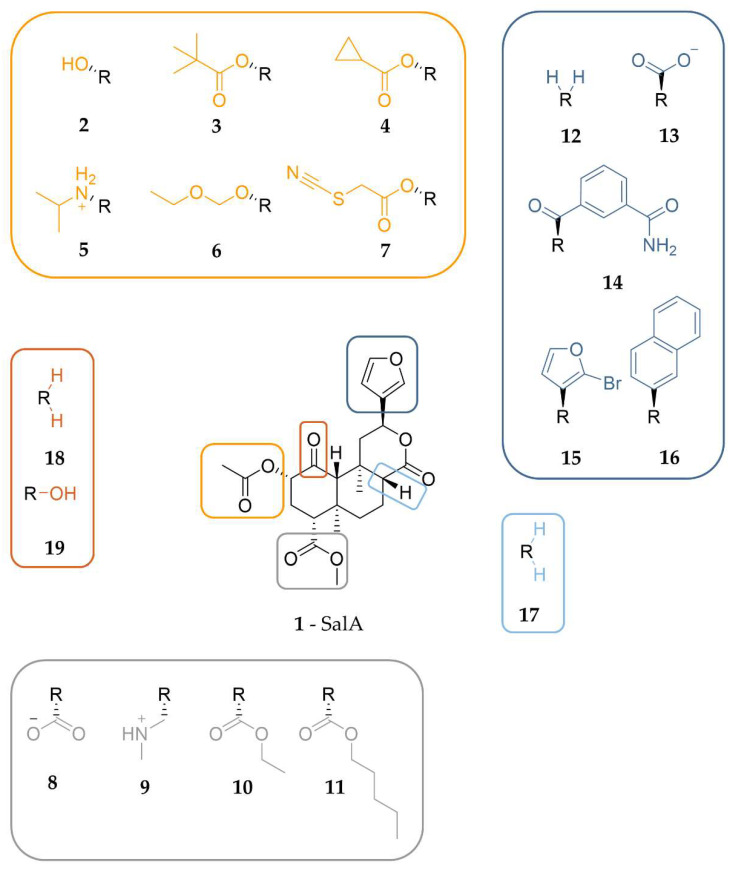
Chemical structure of SalA-analogs discussed to evaluate the proposed binding mode of SalA.

**Table 1 molecules-28-00718-t001:** Protein-ligand interactions between SalA and the KOR.

Interaction	Functional Group	Residue
Hydrogen bond	C1-carbonyl	Y312^7.35^
Hydrogen bond	C2-carbonyl	Y313^7.36^
Hydrogen bond	C4-carbonyl	C210^ECL2^
Hydrophobic contact	C19	V118^2.63^, Y313^7.36^
Hydrophobic contact	C20	I316^7.39^
Hydrophobic contact	Furan	I294^6.55^; Y139^3.33^

**Table 2 molecules-28-00718-t002:** Interactions between SalA and the KOR during MD simulations.

Interaction	Functional Group	Residue	Frequency (%)
Hydrogen bond	C2-carbonyl	Y312^7.35^, Y313^7.36^	17.6
Hydrogen bond	C4-carbonyl	C210^ECL2^	94.1
Hydrogen bond	C1-carbonyl	Y312^7.35^	79.0
Hydrogen bond	C17-carbonyl	Q115^2.60^	54.0
Hydrophobic contact	C19	V118^2.63^, Y313^7.36^	97.3
Hydrophobic contact	C20	I316^7.39^, Y312^7.35^	76.0
Hydrophobic contact	Furan	I135^3.29^, L212^ECL2^; Y139^3.33^	99.5

**Table 3 molecules-28-00718-t003:** Non-conserved residues within the opioid receptor family participating in SalA binding.

Residue	KOR	MOR	DOR	NOP
2.63	V118	N129	K108	D110
6.55	I294	V302	V281	V283
7.35	Y312	W320	L300	L301
7.36	Y313	H321	H301	R302
7.39	I316	I324	I304	T305

**Table 4 molecules-28-00718-t004:** Experimental data for binding of SalA and all analogs discussed within this section measured in radioligand binding assays.

Ligand	Affinity (K_i_) [nM]	K_i_ [ligand]/K_i_ [SalA]	Reference
2 (SalB)	155 ± 23 ^c^	119	[38]
	>10,000 ^a^	-	[72]
	111 ± 12 ^c^	85	[71]
	155 ± 23 ^c^	119	[73]
	280 ± 20 ^d^	147	[47]
	>10,000 ^b^	-	[68]
3	>10,000 ^a^	-	[72]
4	>10,000 ^a^	-	[72]
	>10,000 ^c^	-	[71]
5	17.6 ± 3.1 ^c^	14	[73]
6	0.32 ± 0.02 ^c^	0.133	[69]
	3.13 ± 0.40 ^b^	0.423	[70]
7 (RB-64)	0.59 ± 0.21 ^b^	0.328	[55]
	39 ± 11 ^c^	2	[55]
	0.6 ± 0.2 ^b^	0.097	[68]
8	>10,000 ^c^	-	[74]
	>1000 ^c^	-	[58]
	>1000 ^c^	-	[59]
9	>10,000 ^c^	-	[73]
10	28.5 ± 0.9 ^c^	22	[58]
11	>1000 ^c^	-	[58]
12	3400 ± 150 ^d^	1789	[42]
13	55 ± 23 ^c^	22	[41]
14	40 ± 1 ^b^	5	[75]
15	2.9 ± 0.3 ^c^	1	[41]
	3.0 ± 0.2 ^d^	2	[42]
16	>8000 ^b^	-	[76]
17	6 ± 1 ^b^	2	[77]
18	18 ± 2 ^b^	5	[77]
19	1125 ± 365 ^b^	281	[77]

^a^ K_i_ determined against [^3^H]bremazocine. ^b^ K_i_ determined against [^3^H]U69,593. ^c^ K_i_ determined against [^3^H]diprenorphine. ^d^ K_i_ determined against [^125^I]IOXY.

**Table 5 molecules-28-00718-t005:** Interactions between RB-64 (**7**) and the KOR during MD simulations.

Interaction	Functional Group	Residue	Frequency (%)
Hydrogen bond	C2-carbonyl	Y313^7.36^ (Y312^7.35^)	22.2
Hydrogen bond	C2-thiocyanate	Y313^7.36^	8.6
Hydrogen bond	C4-carbonyl	C210^ECL2^	93.9
Hydrogen bond	C1-carbonyl	Y312^7.35^	80.1
Hydrogen bond	C17-carbonyl	Q115^2.60^	57.4
Hydrophobic contact	C19	V118^2.63^	94.4
Hydrophobic contact	C20	I316^7.39^	51.7
Hydrophobic contact	Furan	I135^3.29^, L212^ECL2^; Y139^3.33^	99.9

## Data Availability

Data is available from the authors upon reasonable request.

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
