# Peer review of "Solving an Old Puzzle: Elucidation and Evaluation of the Binding Mode of Salvinorin A at the Kappa Opioid Receptor"

_molecules, 2023, doi:10.3390/molecules28020718_

Round 1
Reviewer 1 Report
The manuscript was well written, and my opinion it was interesting.
However, I have a little issue if it is possible for the authors to evaluate or calculate the relative binding energy of ligands in MD trajectories so that it would strengthen the result of this manuscript.
An additional comment is the manuscript is quite long, 35 pages so that it could shut down the reader. I request the authors please kindly concise/shorten the manuscript for improving the manuscript.
As such, I would suggest this manuscript as a minor revision.
Reviewer 2 Report
The manuscript by Kristina Puls and Gerhard Wolber focused on investigating the binding mode of Salvinorin A at the kappa opioid receptor is a very standard computational study on the binding mode of a natural compound at a putative molecular target. Authors just apply routinary computational procedures and exhaustively discuss the agreement of their results with several literature data, often quite old. No experimental validations are reported. Nevertheless, these determinations are necessary to demonstrate the binding of the molecule to the receptor, the importance of certain residues over others. I understand that site-directed mutagenesis may have its pitfalls, but it is the only way to demonstrate that a specific residue is involved in the binding with the molecular target. Without any experimental proof, the manuscript is only a set of hypothesis. Thus in the present form it can not be accepted and it should be reinforced by site-directed mutagenesis data or by other experimental proofs demonstrating the interaction of Salvinorin A with specific amino acid residues located on the kappa opioid receptor.
Reviewer 3 Report
In the present paper, the authors computationally investigated the binding of Salvinorin A (SalA) to the kappa opioid receptor (KOR). Despite many previous experimental and theoretical studies of the binding mode of SalA to KOR, the exact structure of the bonded complex remained elusive, primarily because the crystal structure of the active state of KOR was not solved. Unlike the previous studies, in this work the authors employed in the meantime resolved structure of the KOR as a starting point in computational modelling. A combination of molecular docking and MD simulations enabled the identification of ligand poses in the receptor and the analysis of specific interactions of KOR residues with functional groups of SaIA. This comprehensive study also addressed available mutagenesis studies and structure known structure-activity relationship (SAR) data. Overall, this is a good piece of work that is well set out, and my opinion is that it could be published in Molecules journal. The only problem in my opinion is the paper’s length (35 pages!). Since a large amount of space is used for figures, the authors should consider moving some of them to the SI.
